# Up-regulation of plasma miRNA-21 and miRNA-422a in postmenopausal osteoporosis

**Neda Mohammadisima[1], Azizeh Farshbaf-khalili[2]\*, Alireza Ostadrahimi[3]\***

**1** Department of Biochemistry and Dietetics, Faculty of Nutrition and Food Sciences, Tabriz University of Medical Sciences, Tabriz, Iran, **2** Physical Medicine and Rehabilitation Research Center, Aging Research Institute, Tabriz University of Medical Sciences, Tabriz, Iran, **3** Nutrition Research Center, Department of Clinical Nutrition, Tabriz University of Medical Sciences, Tabriz, Iran

\* farshbafa@tbzmed.ac.ir (AFK); ostadrahimi@tbzmed.ac.ir (AO)

## Abstract

### Background

Many researchers focused on diverse miRNAs in the progression of osteoporosis in post-menopausal women. This study aimed to evaluate the association between plasma miRNA-21-5p and miRNA-422a with osteoporosis in postmenopausal women.

### Methods

This cross-sectional comparative study was performed on 126 randomly selected postmenopausal women aged 50–65, including 65 osteoporotic and 61 normal-bone mineral density (BMD) women. miRNA-21 and miRNA-422a were identified using qRT-PCR in these women. BMD was evaluated by the dual-energy X-ray absorptiometry method. A binary logistic regression model adjusted for confounders was used to evaluate the associations between plasma miRNAs' expression levels and osteoporosis. The Area Under Curve (AUC) was calculated to differentiate low BMD in the postmenopausal period using Receiver-Operator Characteristic (ROC) curves.

### Results

miRNA-21 and miRNA-422a were significantly up-regulated in osteoporotic compared to non-osteoporotic postmenopausal women. The expression levels of miRNA-21 and miRNA-422a indicated a significant reverse correlation with both lumbar spine and femoral neck density. After adjusting the confounders, the likelihood of osteoporosis in the postmenopausal women with under the median plasma levels of miRNA-21 (OR = 0.025; 95% CI: 0.003 to 0.198, $p<0.001$) and miRNA-422a (OR = 0.037; 95% CI: 0.007 to 0.211, $p<0.001$) was significantly less than the women with the levels above the median. There were significant inverse correlations between miRNA-21 ($p<0.001$, r = -0.511) and miRNA-422a ($p<0.001$, r = -0.682) with BMD-lumbar spine as well as an inverse correlation between miRNA-21($p<0.001$, r = -0.374) and miRNA-422a ($p<0.001$, r = -0.602) with BMD-femoral neck. The AUC (95%CI) for miRNA-21 and miRNA-422a was 0.84 (0.77 to 0.91) and 0.98 (0.96 to 0.99), respectively. ROC analysis illustrated that sensitivity and specificity values

**Data Availability Statement:** All relevant data are within the paper and its Supporting information files.

**Funding:** The Vice-chancellor for Research and Technology, Tabriz University of Medical Sciences

financially funded the original research (grant no: 61494). The funders had no role in study design, data collection and analysis, decision to publish, or preparation of the manuscript. No any authors received a salary from funder.

**Competing interests:** The authors have declared that no competing interests exist.

were 83.1% and 74%, respectively, for miRNA-21 at the cut-off point of 3.38. Also, at the cut-off point of 2.86, a sensitivity of 94% as well as a specificity of 89% was determined for miRNA-422a.

## Conclusions

This study indicated that the odds of osteoporosis in postmenopausal women increased with the higher expression of plasma miRNA-21 and miRNA-422a.

## Introduction

Osteoporosis is a systemic skeletal disorder characterized by decreased bone mass and impaired bone microarchitecture due to an imbalance between bone formation and resorption during the remodeling cycle [1]. Several factors such as age, genetics, physical activity, smoking, body mass index (BMI), air pollution, stress, vitamin D and calcium deficiency, and other factors affect osteoporosis [2]. Postmenopausal osteoporosis is the most common type of osteoporosis associated with estrogen deficiency during this period [3]. Osteoporosis in postmenopausal women increases the risk of fracture and affects survival. Studies have shown that 50% of women with femoral or pelvic bone fractures are unable to walk again, furthermore, 20% of these women die in less than two years [4].

Currently, dual-energy X-ray absorptiometry (DEXA), bone turnover markers, and FRAX are employed to diagnose osteoporosis or assessment the fracture risk; however, these methods have low specificity and sensitivity. In the past decade, many types of research have focused on circulating miRNAs as a potential biomarker for detecting osteoporosis and fracture [5].

MicroRNAs (miRNAs) are non-encoded, endogenous, and single-stranded RNA molecules consisting of about 22 nucleotides that regulate biological processes by binding to 3'-UTR in mRNA [6, 7].

In vivo and in vitro evidence suggests that miRNAs are involved in cell proliferation, differentiation, survival, and apoptosis of bone mesenchyme stem cells (BMSCs), osteoblasts, osteoclasts, osteocytes, adipocytes, and chondrocytes [8]. Many researchers focused on diverse miRNAs in the progression of osteoporosis in postmenopausal women. These studies indicated up and down-regulation of different miRNAs in postmenopausal osteoporotic versus postmenopausal healthy women [9].

Among these, miR-422a stimulates osteoclast genesis by binding to 3'-UTR mRNA regions of PAG1, CD22, CBL, TOB, and IGF1 genes, so reduces BMD in postmenopausal women.[10] Studies regarding the association of miRNA-422a with osteoporosis suggested that miR-422a might be increased in osteoporotic postmenopausal women compared to healthy postmenopausal women [10, 11]. However, there are few studies on circulating miR-422a in postmenopausal osteoporosis Expression levels of miR-21-5p increase in osteoporosis and bone fractures [12–14]. In addition, some studies reported a decrease of miRNA-21 in osteoporosis patients compared to the normal bone mineral density group [11, 15]. In addition, the conclusions of studies on circulating miRNA-21 in osteoporosis are inconsistent. Further studies are needed to elucidate the role of these miRNAs in postmenopausal osteoporosis.

This study aimed to evaluate the plasma expression level of miRNA-21 and miR-422a in osteoporotic postmenopausal women compared to those with normal BMD.

## Method and materials

### Participants

This is a cross-sectional comparative study taken from a Megaproject [16]. All procedures involving people comply with the ethical standards of the institutional and/or national committee for research ethics and the 1964 Helsinki Declaration and its subsequent changes or comparable ethical standards. The Ethics Committee of Tabriz University of Medical Sciences (IR.TBZMED.REC.1398.215) approved the protocol of this study.

First, out of 87 health centers in Tabriz, using the integrated health system, women aged 50–65 years were selected and numbered consecutively in a list from number one to the end. Out of 108778 women, 850 postmenopausal women were selected by simple random sampling. By telephone interview, 730 of them were eligible to participate in the study and were asked to attend health centers. Informed voluntary written consent was obtained from each of the participants. After checking the inclusion criteria by the checklist, 194 women were excluded from the study due to a lack of inclusion criteria. Reasons for exclusion were: consumption of corticosteroids (n = 37), fracture history (n = 23), menopause earlier than 40 years (n = 17), hyperthyroidism (n = 30), rheumatoid arthritis (n = 15), malignancy (n = 14), kidney failure or disease (n = 4), oral bisphosphonate consumption in the last 6 months (n = 6), chronic liver disease (n = 3), hyperparathyroidism (n = 4), phenytoin consumption (n = 7), reluctance to continue study (n = 34).

Out of the remaining 536 women, blood samples were collected for CBC/diff, calcium, phosphorus, alkaline phosphatase, TSH, creatinine, blood glucose, and 25-hydroxyvitamin D tests to differentiate primary osteoporosis from secondary osteoporosis. According to the results, 74 women were identified and excluded due to secondary osteoporosis. Seventeen women were excluded from the study due to their unwillingness to continue participation. Finally, demographic, midwifery, anthropometric, physical activity, and food frequency questionnaires were completed for 445 postmenopausal women.

They were referred to the Bone Densitometry Center. Based on densitometric tests, 142 normal-BMD, 109 osteoporosis, and 194 osteopenia women were identified. Of these, plasma samples of osteoporotic and norma-BMD women with hypertension and heart disease were excluded. Finally, 61 patients in the normal-BMD group and 65 patients in the osteoporosis group remained and were examined for plasma miRNA (Fig 1).

### Sample size

The sample size was calculated using G*POWER software (version 3.1.2) according to the plasma levels of miRNA-21 in the Li et al. study [15]. Based on the correlation coefficient between miRNA-21 with lumbar vertebral density in normal BMD (r = 0.457) and osteoporotic (r = 0.466) subjects; also based on the correlation coefficient between miRNA-21 with hip bone density in normal BMD (r = 0.482) and osteoporotic (r = 0.464) women using a two-tailed test, 95% confidence level, power of 90%, the sample size was calculated 49 and 50 participants for osteoporotic group and 47 and 52 participants for the normal-BMD group. Taking into account the 20% probable dropout, the final sample size was estimated as 60 participants for each group.

### Measurements

Demographic characteristics and physical activity questionnaires were obtained from women. The weight of participants was measured using digital scales (Seca, Germany). Height was measured using a wall-mounted stadiometer (Seca, Germany). By dividing weight in

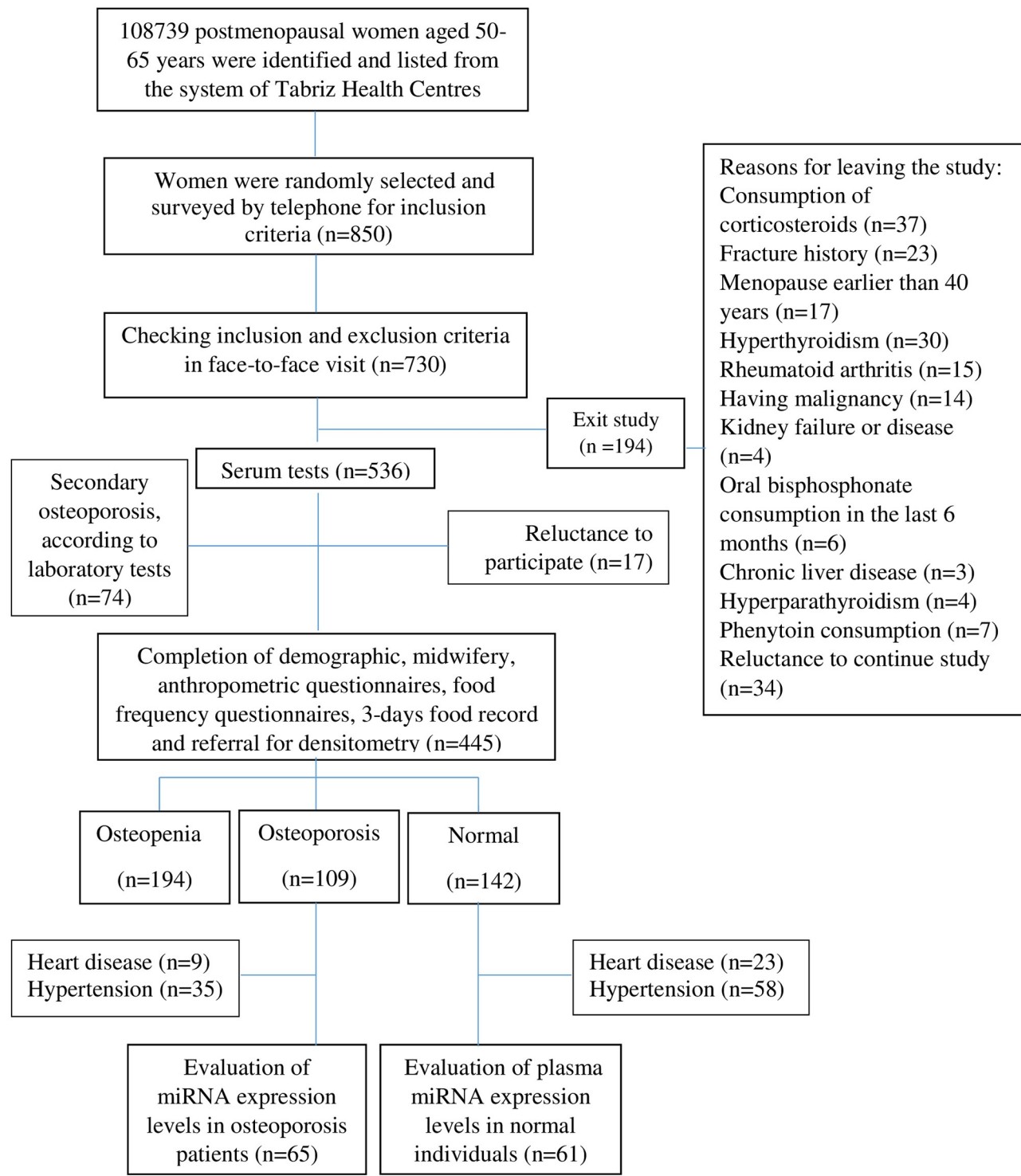

**Fig 1. Flowchart of the study population selection.**

kilograms by the square of the height in meters (kg/m$^2$), BMI was calculated. For the evaluation of physical activity, the valid and reliable International Physical Activity Questionnaire-Short Form, which has been approved in Iran [17] was completed and total physical activity was calculated. Dietary intake was evaluated by a reliable and valid semi-quantitative food frequency questionnaire during the past year [18]. The method of its calculation has been mentioned elsewhere [19]. The BMD was obtained based on measurements in the femoral neck and lumbar spine (L1-L4) using a dual-energy X-ray absorptiometry machine (Hologic QDR 4500W (S/N 50266) (DEXA) and reported by T-score criteria. According to the World Health Organization, T-scores higher than –1.0 are classified as normal, and T-scores lower than –2.5 are classified as osteoporosis [20]. An endocrinologist did the final diagnosis of osteoporosis.

## Plasma sample collection

Five cc blood samples were taken from each patient after 10–12 hours of overnight fasting and were transferred to a sterile tube containing anticoagulants EDTA (BD Vacationer; Becton Dickinson and Company, Franklin Lakes, NJ). The collected blood sample was sent to the laboratory in less than an hour. Plasma was separated by centrifugation at 2,000×g for 20 minutes at 4˚C. The supernatant was transferred to a sterile tube. Extracted plasma samples were stored at -80˚C.

## RNA extraction

RNAs were extracted from plasma using TRIzol® LS reagent (Thermo Fisher Scientific, Korea). First, 700 µL of TRIzol® LS Reagent (Thermo Fisher Scientific, Korea) was added to the 200 µL plasma sample. After homogenization, the solution was transferred to the microtube, and then about 200 µL of chloroform was added and allowed to stand on ice for 10 minutes. The tube was then gently inverted for 15 seconds. The tube containing the sample was re-incubated in ice for 15 min. The tube was centrifuged at 12000×g for 15 minutes at 4˚C using a SIGMA centrifuge. After centrifugation, the solution in the tube was divided into three phases: the upper phase containing RNA (clear blue), the middle phase containing DNA (white), and the lower phase containing protein and some DNA (purple). The supernatant was transferred to a new RNase-free microtubule, to which 1 ml of cold isopropanol was added. Incubation was performed in the freezer for 20 minutes. Then, the desired tube was centrifuged with the previous conditions (i.e. 12000×g and 4˚C) for 15 minutes. After centrifugation, the RNA precipitated and appeared as a plate on the bottom of the microtube. First, all supernatants were removed, and then 1 ml of ethanol 75% was added to the RNA precipitate. It was centrifuged at 7500×g for 10 minutes at 4˚C. After centrifugation, first, the supernatant was removed, and then the plate was completely air-dried. Then about 40 µL of DEPC (diethyl pyro carbonate) water solution was added. Using the Dry Bath, the precipitated RNA was placed at 60˚C for 15 minutes to dissolve in water. After rotation, the RNA concentration of samples was measured using a NanoDrop ND-1000 Spectrophotometer (Thermo Fisher), and to estimate the purity of RNA, The A260/A280 and A260/A230 were obtained. The A260/A280 ranged from 1.8 to 2.2, indicating pure RNA. The RNAs isolated were immediately stored in a −80˚C freezer.

## Complimentary DNA (C-DNA) synthesis

After determining the concentration and purity of the extracted RNAs and ensuring the absence of phenolic, protein, and DNA contamination in the extracted solution, cDNA was synthesized using Prime Script cDNA synthesis kits (Takara Bio, Japan). We mixed the appropriate volume of RNAs with the appropriate amount of RNase-Free water so that the final concentration of RNA in the samples reaches 3 µg/µL. In the first step, RNase-Free water was

added to a proper volume of extracted RNA. Then, one μL of miRNAs-specific primers and one μL primer of U6 (as an internal control) were added. The final volume with RnaseFree water reached 13μL. The above compound was then incubated for 5 minutes at 65°C. The reason for this incubation was to open the secondary structures in the RNA structure, single-strand it, and attach the primer to the template strand. One μl Reverse Transcriptase, 4μl of buffer for cDNA synthesis, was added to the above mixture. The final volume reached 20μL. To remove the bubbles and perform PCR correctly, the samples were spun, and cDNAs were synthesized by using a Thermal Cycler (Bio-Rad) according to the following conditions: for 5 minutes at 25°C and 30 minutes at 37°C. The temperature was raised to 85°C for 5 seconds and finally reached 4°C.

## Real-time quantitative PCR (qRT-PCR)

To evaluate the expression of each gene according to the number of samples PCR reaction was used in an SYBR Premix Ex Taq Kit (Takara Bio, Japan) at a final volume of 10μL. First, a mixture of 5μL SYBR® Green PCR Master MixTaq (2x), 0.3μL Mix Primer forward and reverse (The sequence of the primers used is shown in Table 1), and 3.7μL DEPC-treated water was prepared and after distributing nine μL of the mixture in special tubes, 1μL of the synthesized cDNA was added to each. Then the tubes were closed, and the samples were run on Step One Plus Real-Time PCR (Applied Biosystems, USA) instrument.

Temperature conditions and real-time cycles for the cDNA and u6 genes were as follows: 94°C for 3 min, 94°C for 10 seconds, 59° C for 30 seconds, and 72° C for 20 seconds at 50 cycles. The cycle of threshold (Ct) indicates the expression level of plasma miRNAs and U6 (as internal control). The amplification curves were analyzed for the determination of the cycle of threshold (Ct). The Ct value normalized with U6, using the $2^{-\Delta\Delta CT}$ (fold change) method in the analysis of real-time PCR as previously described [21]. In addition, a melting curve was drawn to ensure the accuracy of the PCR.

## Statistical analysis

All the statistical analysis was performed using SPSS version 24.0. A p-value less than 0.05 was considered a significance level. The normality of continuous data among groups was evaluated by Kolmogorov-Smirnov and dispersion indicators (SD, Skewness-kurtosis test). For determining differences between osteoporosis and healthy women, we used an independent t-test for continuous data by a normal distribution and the Mann-Whitney test for data by abnormal distribution. Logarithmic transformation (log10 plus 1.5 fold change) was used to normalize the expressed miRNAs. Pearson correlations were used to evaluate the correlation between miRNAs and BMD at the lumbar spine & femoral neck. The binary logistic regression model was used to estimate the odds ratio (95% confidence intervals) of osteoporosis based on miR-NAs as a continuous and dichotomous variable according to the median of the miRNA for the

**Table 1. The primer sequence used for qRT-PCR.**

| | | primer |
|---|---|---|
| hsa-miRNA-21-5p | Forward | 5'–CCGCAGGTAGCTTATCAGA–3' |
| | Reverse | 5'–ATGGAGCCTGGGACGTGACC–3' |
| hsa-miRNA-422a-5p | Forward | 5'–CCGCAGGACTGGACTTAGG–3' |
| | Reverse | 5'–ATGGAGCCTGGGACGTGACC–3' |
| U6 (Reference miRNA) | Forward | 5'–TAAAATCATATACACGACGGCTTCG–3' |
| | Reverse | 5'–TACTGTGCGTTTAAGCACTTCGC–3' |

control group adjusted for age, BMI, and DII. To determine the sensitivity and specificity of threshold values for miRNA-21 and miRNA-422a to differentiate low bone density in the early postmenopausal period (50 to 65 years), we applied receiver-operator characteristic curves (ROC).

## Results

### Demographic characteristics

Characteristics of participants in the normal-BMD and osteoporotic postmenopausal women are demonstrated in Table 2. Significant differences were found for age, BMI, DII, lumbar spine, and femoral neck BMD, T-score, and Z-score in the osteoporosis and normal-BMD postmenopausal women.

### Association of expression levels of miRNAs with osteoporosis

Table 3 indicates the mean (SD) expression level of miRNA-21 and miRNA422a in the normal-BMD and osteoporotic postmenopausal women and odds ratios (OR) with 95% confidence intervals (CI) of osteoporosis in postmenopausal women based on the expression level of miRNAs adjusted for age, BMI and DII (reference group: normal-BMD). The mean plasma expression level of miRNA-21, and miRNA-422a was significantly higher in the osteoporotic

**Table 2. Characteristics of participants in the normal-BMD and osteoporosis postmenopausal women.**

| Mean± SD or N (%) | Osteoporosis (n = 65) | Normal-BMD (n = 61) | *p*-value |
|---|---|---|---|
| Age (years) | 58.2 ± 3.8 | 55.7 ± 3.6 | <0.001[a] |
| Menopause age (years) | 48.3 ± 3.9 | 49.4 ± 3.9 | 0.14[a] |
| Body Mass Index (BMI) kg/m2 | 28.1 ± 4.0 | 31.9 ± 5.0 | 0.001[a] |
| Married | 49 (75.4) | 53 (86.9) | 0.078[ch] |
| History of injection contraceptive | 6 (9.5) | 4 (7.3) | 0.661[ch] |
| Energy (Kcal/day) | 1786.3 (438.5) | 1905.2 (419.4) | 0.13[a] |
| Total Met [d] (Mets- min/week) | 346.5 (959.7) | 401.3 (673.5) | 0.84[c] |
| History of smoking or smoker roommate | 21 (32.7) | 21 (36.7) | 0.700 [b] |
| Vitamin D3 (ng/ml) | 44.73 ± 20.65 | 48 ± 22.93 | 0.414[a] |
| Income | | | 0.400[ch] |
| Adequate | 18 (29.5) | 13 (20.3) | |
| Some adequate | 36 (59.0) | 40 (62.5) | |
| Never adequate | 11 (17.2) | 7 (11.5) | |
| DII [d] | 0.91 (1.87) | -0.04 (2.03) | <0.001[c] |
| L.S BMD (g/cm$^2$) | 0.71 ± 0.07 | 1.04 ± 0.11 | <0.001[a] |
| L.S T-score | -3.1 ± 0.95 | 0.02 ± 0.59 | <0.001[a] |
| L.S Z-score | -1.76 ± 0.63 | 1.15 ± 0.98 | <0.001[a] |
| F.N BMD (g/cm$^2$) | 0.74 ± 0.95 | 0.99 ± 0.13 | <0.001[a] |
| F.N T-score | -1.67 ± 0.78 | 0.29 ± 0.86 | <0.001[a] |
| F.N Z-score | -0.76 ± 0.83 | 1.06 ± 0.86 | <0.001[a] |

L.S: lumbar spine, F.N: femoral neck, BMD: bone mineral density, DII: dietary inflammatory index.

[a]: Independent t-test,

[b]: Fisher's Exact Test,

[c]: Mann Whitney, ch: Chi-square,

[d]: Median (IQR)

**Table 3. The plasma expression level of microRNA and the odds of osteoporosis according to microRNA expression level according to the binary logistic regression model in postmenopausal women\*.**

| miRNA* | Osteoporosis (n = 65) | | Normal-BMD (n = 61) | *p*-value |
|---|---|---|---|---|
| | Mean± SD | | Mean± SD | |
| miRNA-21 | 1.03 ± 0.36 | | 0.58 ± 0.26 | <0.001[€] |
| miRNA-422a | 2.71 ± 0.43 | | 1.42 ± 0.47 | <0.001[€] |
| | miRNA (Continuous) OR (95% CI) | P-value | miRNA (categorical)[e] OR (95% CI) | P-value |
| miRNA-21 | 1.67 (1.180–2.363)[a] | 0.004[±] | 0.025 (0.003–0.198)[c] | <0.001[±] |
| miRNA-422a | 1.494 (1.098–2.034)[b] | 0.011[±] | 0.037 (0.007–0.211)[d] | <0.001[±] |

\* Log10 plus 1.5 fold change of miRNA;

[€]Independent t-test

[±]Adjusted for age, BMI, and DII

[a] Hosmer and Lemeshow p = 0.602, Chi-square = 6.403, df = 8

[b] Hosmer and Lemeshow p = 0.977, Chi-square = 2.133, df = 8

[c] According to the median of the miRNA for the controls (0.717); Hosmer and Lemeshow p = 0.238, Chi-square = 10.4, df = 8

[d] According to the median of the miRNA for the controls (2.075); Hosmer and Lemeshow p = 0.369, Chi-square = 8.69, df = 8

[e] Reference group: normal- BMD group

compared to normal-BMD women. The results of the binary logistic model showed the higher levels of miRNAs associated with higher odds of having osteoporosis from modeling miRNA-21 (OR = 1.67; 95% CI = 1.180–2.363, *p* = 0.004) and miRNA-422a (OR = 1.494; 95% CI = 1.098–2.034, *p* = 0.011) as continuous variables. Also, we observed significant associations between miRNA-21 and miRNA-422a with osteoporosis after adjusting age, BMI and DII obtained from modeling miRNA-21 as a dichotomous variable based on the median (0.717) of miRNA-21 (OR = 0.025; 95% CI = 0.003–0.198, *p*<0.001), and modeling miRNA-422a as a dichotomous variable based on the median (2.075) of miRNA-422a (OR = 0.037; 95% CI = 0.007–0.211, *p*<0.001). The probability of osteoporosis was significantly lower in the women with the miRNA-21 and miRNA-422a expression levels under the median compared to the other group.

Fig 2 shows a significant inverse correlation between miRNA-21 (*p*<0.001, r = -0.511) and miRNA-422a (*p*<0.001, r = -0.682) with BMD-lumbar spine as well as an inverse correlation between miRNA-21(*p*<0.001, r = -0.374) and miRNA-422a (*p*<0.001, r = -0.602) with BMD-femoral neck.

### Diagnostic accuracy of BTMs

To investigate the diagnostic accuracy of the plasma level of miRNA-21 and miRNA-422a compared to the DXA standard method for low-BMD, the ROC curve was used [Fig 3]. AUC: 95% confidence interval, standard error, and p-value for miRNA-21 and miRNA-422a in comparison to DXA were obtained 0.84: 0.77 to 0.91, 0.036; p<0.001 and 0.98: 0.96 to 0.99, 0.01; p<0.001], respectively.

A sensitivity of 83.1%, specificity of 74%, positive likelihood ratio of -3.17, negative likelihood ratio of -0.23, a positive diagnostic value of 78%, and negative diagnostic value of 81%, were attained for miRNA-21 at the cut-off point of 3.38. At the cut-off point of 2.86, a sensitivity of 94%, specificity of 89%, positive likelihood ratio of -8.16, negative likelihood ratio of -0.070, positive diagnostic value of 90%, and negative diagnostic value of 93% were obtained for miRNA-422a.

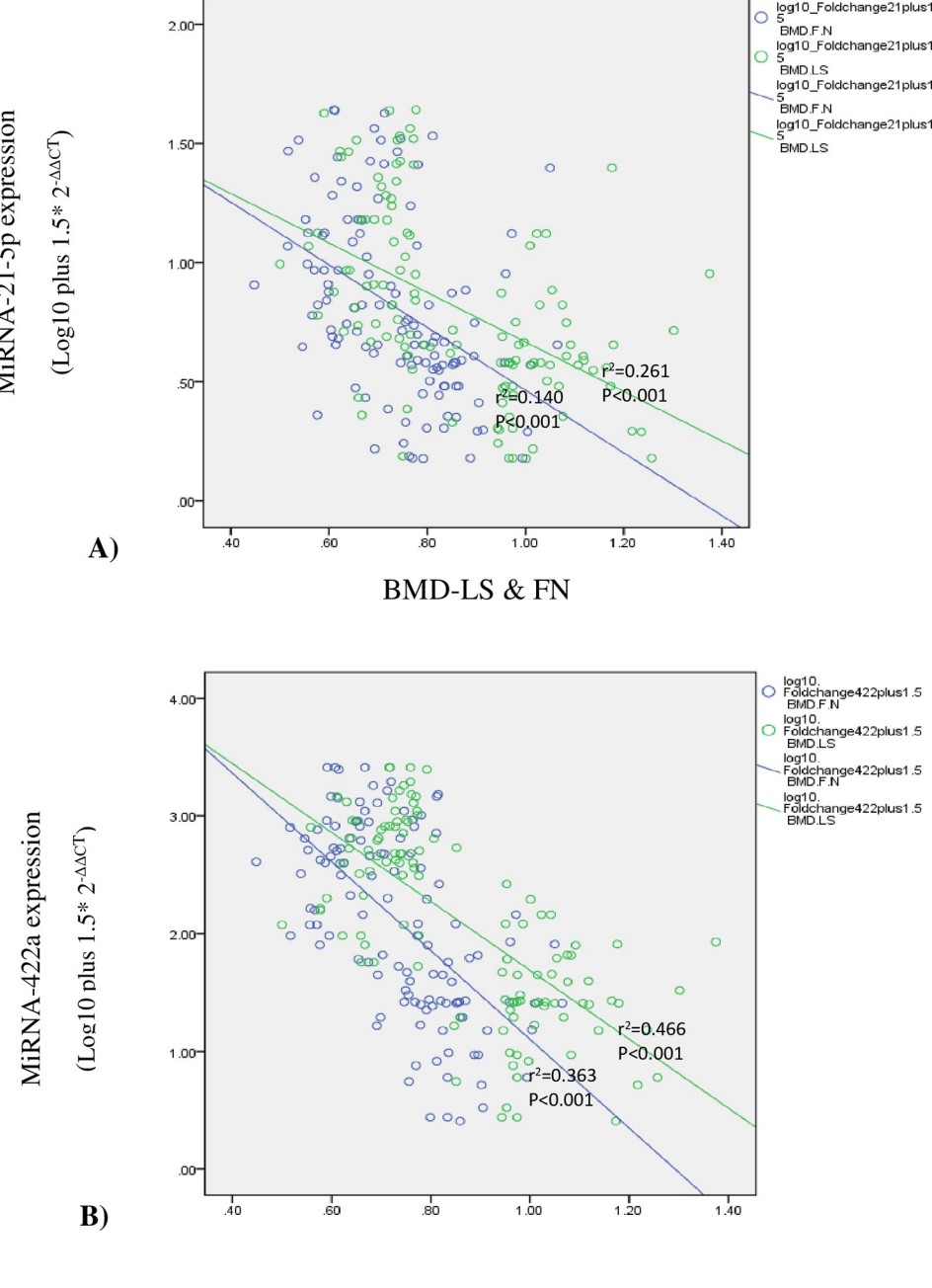

**Fig 2. Correlation between miRNA-21 and miRNA-422a with BMD at lumbar spine & femoral neck (all p<0.001) (Part A and B).**

## Discussion

This study demonstrated the odds of osteoporosis increased in postmenopausal women with the higher expression level of miRNA-21 and miRNA-422a.

miRNAs are exported from cultured cells into the extracellular space and are detectable in diverse biofluids, including plasma, serum, urine, saliva, etc [22]. Plasma is preferable to serum

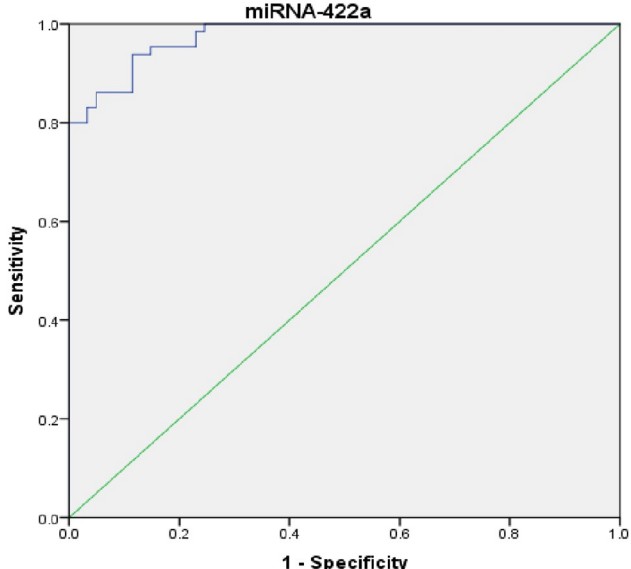

**A)**

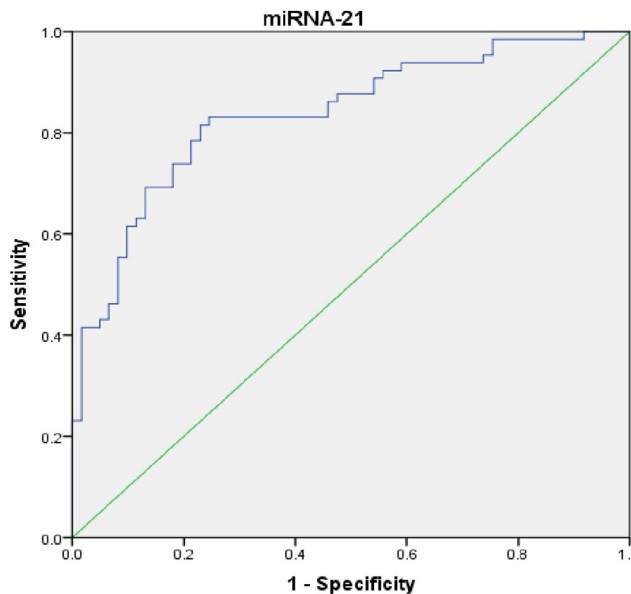

**B)**

**Fig 3. Receiver operating characteristic (ROC) curves for miR-21 and miR-422a (Part A and B).**

for the measurements of miRNA; although, in human studies, the difference was not significant [23]. miRNA-21 upregulated in a variety of cancers [24], and heart disease [25, 26].

miRNA-21 is involved in both osteoclast genes [27] and osteoblast genes [28]. C-Fos osteoporotic transcription factor increases miRNA-21 expression. The miR-21 then inhibits

PDCD4 (programmed cell death protein) expression. Lowering PDCD4 stimulates c-Fos activation, which leads to the expression of NFATc1 and the differentiation of bone marrow-derived monocyte/macrophage precursors (BMM) into osteoclasts. As a result, miRNA-21 increases RANKL-induced osteoporosis [29] (S1 Fig). MiRNA-21 inhibits osteogenesis by regulating exosomes taken from MSCs in osteoporotic patients by targeting SMAD7 [30]. On the other hand, miRNA-21 promotes osteogenesis by targeting Smad7-Smad1/5/8-Runx2 [31], PI3K/β-catenin [32], and PTEN/PI3K/Akt/HIF-1α pathways [33] (S2 Fig). Zhao et al. indicated miRNA-21 modulated osteoporosis by decreasing RECK in the OVX mice [34]. A recent study assessed the dual role of miRNA-21 in osteoblast-osteoclast coupling. This study demonstrated that inhibiting miRNA-21 in Pre-osteoblastic mouse cell line MC3T3, increased the expression of genes encoding osteogenic markers, including osteocalcin, collagen type I, and Runx-2. In addition, inhibition of miRNA-21 inflected the synthesis of OPN protein in MC3T3 culture, which is necessary for proper extracellular matrix mineralization and anchoring of osteoclasts to the bones. Osteoplastic miRNA-21 regulates factors that are essential for pre-osteoclasts, such as RANKL. Furthermore, inhibition of miR-21 in the pre-osteoclast cultured cells leads to a decrease in the expression of markers related to the differentiation of osteoclasts such as RANKL, and cathepsin K [35]. Suarjana et al. observed higher serum miRNA-21 expression in postmenopausal women with osteoporosis compared to non-osteoporotic postmenopausal women. Low estrogen levels in postmenopausal women increased serum miRNA-21 expression. The excessive expression of miRNA-21 increases (the receptor activator of NF-kB ligand (RANKL) and decreases osteoprotegerin (OPG) and Transforming growth factor-beta (TGF-β). Increased levels of RANKL and subsequently decreased levels of OPG lead to an increase in the RANKL/OPG ratio, and eventually to a decrease in BMD [13]. Estrogen downregulates miRNA-21, which causes up-regulate transcription of the FasL gene in osteoclasts. Enhanced-FasL proteins stimulate caspase-3 activity, so increase osteoclast apoptosis [26]. The results of our study agree with the results of the Suarjana et al. study. However, estrogen levels in postmenopausal women have not been evaluated in our study. In another study, the downregulation of miRNA-21 in the plasma of osteoporosis and osteopenia women against the normal group was demonstrated [15]. The result of our study contradicts this study. Women with secondary osteoporosis were excluded from our study, but they did not exclude them. Postmenopausal women were also not evaluated for other diseases, such as malignancy and heart disease in their study, which may affect miRNA levels. Another possible explanation could be the difference between ethnicity (Iranian vs. Chinese).

miRNA-422a plays important role in the development of several types of cancers such as esophageal squamous cell carcinoma [36] and lung cancer [37]. On the other hand, miRNA-422a is a tumor suppressor in colorectal cancer [38] and breast cancer [39]. There are insufficient studies on the association between miRNA-422a and osteoporosis; however, these agree with the results of our study. Cao et al. indicated that miR-422a might regulate osteoporosis in postmenopausal women and may be a potential marker for osteoporosis [10]. A cross-sectional study found the expression of miRNA-21 was lower in the osteoporosis group than in the control group. Moreover, the expression of miRNA-422a in the osteoporosis group was higher than in the normal group, although the difference was not statistically significant [11]. The obvious difference between this study and ours is the comparison of osteoporotic women with and without bone fractures versus healthy women. In addition, 90% of osteoporosis women who participated in the study were taking anti-osteoporosis drugs, which may affect miRNA levels.

Our study was a random selection of participants from different geographic areas with diverse socioeconomic statuses. These participants were screened for secondary osteoporosis and so many diseases, which may affect the expression level of miRNA were excluded. These

diseases included Rheumatoid arthritis, malignancy, kidney failure or disease, chronic liver disease, heart disease, and hypertension. In addition, age, BMI, and DII were adjusted as confounding factors for the comparison of postmenopausal women with and without osteoporosis. However, there may be other unknown or known factors such as infectious, mental diseases and lifestyle affect miRNAs. Blood sampling was performed after an overnight fast between 8:00 a.m. and 9:30 a.m. This approach was implemented concerning the fact that miRNAs may be affected by the circadian rhythm [40]. The present study includes an age limit of 50–65 years for participants. Although in this research, it was possible to use only two miRNAs and one internal control gene, we suggest using several miRNAs and internal control for future studies. The sample size in this study was small, so conducting future studies with a larger sample size is recommended. miRNA-21-5p and miRNA-422a can be predicted through bioinformatic analysis and subsequently verified.

## Conclusion

The present study indicated that with the increased expression of plasma miRNA-21 and miR-422a, the odds of having osteoporosis increased in postmenopausal women. In addition, we observed the significant inverse correlation between miRNA-21 and miRNA-422a with BMD-lumbar spine and BMD-femoral neck. miRNA-21 and miRNA-422a had good and excellent diagnostic accuracy, respectively. So, these miRNAs' expression levels recommend as helpful markers for screening women at risk of osteoporosis.

## Supporting information

**S1 Fig. MiRNA-21 in osteoclastogenes.**
(TIF)

**S2 Fig. MiRNA-21 in osteoblastogenes.**
(TIF)

**S1 File. Accessible data.**
(SAV)

## Acknowledgments

We appreciate the headquarters and personnel of the School of Nutrition, the Physical Medicine and Rehabilitation Research Center, the Bone Density Department of Sinai Hospital, and all women who patiently participated in this research.

## Author Contributions

**Conceptualization:** Azizeh Farshbaf-khalili, Alireza Ostadrahimi.

**Data curation:** Neda Mohammadisima, Azizeh Farshbaf-khalili.

**Formal analysis:** Neda Mohammadisima.

**Investigation:** Neda Mohammadisima, Azizeh Farshbaf-khalili, Alireza Ostadrahimi.

**Project administration:** Azizeh Farshbaf-khalili.

**Supervision:** Alireza Ostadrahimi.

**Writing – original draft:** Neda Mohammadisima.

**Writing – review & editing:** Azizeh Farshbaf-khalili, Alireza Ostadrahimi.

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
