## [Decision Letter · Decision Letter 0]

10 Apr 2023

PONE-D-22-34068Up-regulation of plasma miRNA-21 and miRNA-422a in postmenopausal osteoporosisPLOS ONE

Dear Dr. Farshbaf-Khalili,

Thank you for submitting your manuscript to PLOS ONE. After careful consideration, we feel that it has merit but does not fully meet PLOS ONE’s publication criteria as it currently stands. Therefore, we invite you to submit a revised version of the manuscript that addresses the points raised during the review process.

We look forward to receiving your revised manuscript.

Kind regards,

Gary S. Stein

Academic Editor

PLOS ONE

Journal Requirements:

Reviewers' comments:

Reviewer's Responses to Questions

**Comments to the Author**

1. Is the manuscript technically sound, and do the data support the conclusions?

Reviewer #1: Yes

Reviewer #2: Yes

2. Has the statistical analysis been performed appropriately and rigorously? 

Reviewer #1: Yes

Reviewer #2: Yes

3. Have the authors made all data underlying the findings in their manuscript fully available?

Reviewer #1: Yes

Reviewer #2: Yes

4. Is the manuscript presented in an intelligible fashion and written in standard English?

Reviewer #1: Yes

Reviewer #2: Yes

5. Review Comments to the Author

Reviewer #1: The research is to verify the association between plasma miRNA-21-5p and miRNA-422a with osteoporosis in postmenopausal women. Certain questions have yet to be clarified.

1. Though miRNAs as biomarkers of postmenopausal osteoporosis have been studied intensively, certain miRNAs expressed in plasma at different stage of postmenopausal osteoporosis are still not clear. In current study, plasma miRNA-21-5p and miRNA-422a were chosen to verify their association with osteoporosis in postmenopausal women. Conclusion: “these miRNAs' expression levels recommend as helpful markers for screening women at risk of osteoporosis.” Which is more helpful or both?

Method and Materials Participants: “They were referred to the Bone Densitometry center. Based on densitometric tests, 142 normal- 85 BMD, 109 osteoporosis, and 194 osteopenia women were identified.”

It is suggested that the expression of miRNA-21-5p and miRNA-422a should be detected in some women qualified the inclusion criteria of 194 osteopenia women. Thus, the candidate miRNAs as biomarkers of postmenopausal patients in the early stage may be further explored through the comparison of osteoporosis, osteopenia, and normal groups, and maybe contribute to predict the occurrence of postmenopausal osteoporosis.

2. Potential target genes of miRNA-21-5p and miRNA-422a can be predicted through bioinformatic analysis and subsequently verified.

3. The primer sequences used for qRT-PCR should be listed as attached table.

4. Please standardize the statistical symbol p as lowercase and italic type.

In general, the current research seems to be not deep enough.

Reviewer #2: In this manuscript, the authors have investigated the association between plasma miRNA-21-5p and miRNA-422a with osteoporosis in postmenopausal women. By evaluating 65 osteoporotic and 61 normal cases of women aged 50-65, it has been concluded that miRNA-21 and miRNA-422 are significantly upregulated in osteoporotic women compared to normal. The results further reveal that there are inverse correlations between the levels of miRNA-21 and miRNA-422a and bone mineral density of spine and femoral neck in osteoporotic samples. The role of miRNA-422a and miRNA21 have been investigated in postmenopausal osteoporosis. For example, miRNA 422a has been studied in osteoclast precursor cells as a biomarker and miRNA 21 has been shown to regulate bone turn over markers. This investigation using plasma miRNAs further confirms that miRNAs 422a and 21 could be explored as diagnostic markers in postmenopausal osteoporosis.

Minor comment:

1.Although, the results are quite convincing and interesting, the sample size (65 osteoporotic and 61 normal) is too small.

2. The manuscript has few typographical errors that need to be fixed.

6. PLOS authors have the option to publish the peer review history of their article (what does this mean?). If published, this will include your full peer review and any attached files.

Reviewer #1: No

Reviewer #2: No

---

## [Author Response · Author response to Decision Letter 0]

26 Apr 2023

Title: Up-regulation of plasma miRNA-21 and miRNA-422a in postmenopausal osteoporosis

Dear Editor-in-Chief,

I hope you are well and thank a lot for your attention to our submitted manuscript. Following are our responses to the Editor in Chief’s and reviewers’ comments. I hope that our corrections will be satisfactory for the respected editor.

Kind regards

PONE-D-22-34068R1

We've checked your submission and before we can proceed, we need you to address the following issues:

1. Thank you for stating the following financial disclosure:

 [No-The funders had no role in study design, data collection and analysis, decision to publish, or preparation of the manuscript.]

Response: The Vice-chancellor for Research and Technology, Tabriz University of Medical Sciences financially funded the original research (grant no: 61494).

Response: It was written. The funders had no role in your study, please state: “The funders had no role in study design, data collection and analysis, decision to publish, or preparation of the manuscript.

Response: No any authors received a salary from the funder.

Response: The authors received no specific funding for this work.

Response: We included our amended statements within the cover letter.

2. Please note that funding information should not appear in the Acknowledgments section or other areas of your manuscript. We will only publish funding information present in the Funding Statement section of the online submission form. Please remove any funding-related text from the manuscript.

Response: We removed it.

3. Please include a separate legend for each figure in your manuscript.

Response: The legend for figures 1, 2, and 3 was included. Figures 2 and 3 comprise parts A and B which were clarified.

4. Please upload a Response to Reviewers letter which should include a point-by-point response to each of the points made by the Editor and/or Reviewers. (This should be uploaded as a 'Response to Reviewers' file type.) Please follow this link for more information: http://blogs.PLOS.org/everyone/2011/05/10/how-to-submit-your-revised-manuscript/

Response: We uploaded a Response to Reviewers letter which should include a point-by-point response to each of the points made by the dear Editor and/or Reviewers.

5. We note that you have indicated that data from this study are available upon request. PLOS only allows data to be available upon request if there are legal or ethical restrictions on sharing data publicly ( http://journals.plos.org/plosone/s/data-availability#loc-unacceptable-data-access-restrictions).

In line with our goal of ensuring long-term data availability to all interested researchers, PLOS’ Data Policy also states that authors cannot be the sole named individuals responsible for ensuring data access ( http://journals.plos.org/plosone/s/data-availability#loc-acceptable-data-sharing-methods).

a) If there are ethical or legal restrictions on sharing a de-identified data set, please explain them in detail (e.g., data contain potentially identifying or sensitive patient information) and who has imposed them (e.g., a Research Ethics Committee or Institutional Review Board, etc.). Please provide non-author contact information (phone/email/hyperlink) for a data access committee, ethics committee, or other institutional body to which data requests may be sent. If applicable, please also provide any necessary information which interested researchers would need when requesting access to data in order to obtain the minimal data set for your study.

Response: We have no restrictions.

b) If your minimal data were obtained from a third party (i.e., data not owned or collected by the authors), please explain how others can access or request the specific datasets related to your research. Confirm that others would be able to access or request these data in the same manner as the authors. Please also confirm that the authors did not have any special access or request privileges that others would not have.

Response: We have no restrictions.

c) If there are no restrictions, and your minimal data were not obtained from a third party, please upload the minimal anonymized data set necessary to replicate your study findings to a stable, public repository and provide us with the relevant URLs, DOIs, or accession numbers. Please see http://www.bmj.com/content/340/bmj.c181.long for guidelines on how to de-identify and prepare clinical data for publication. For a list of recommended repositories, please see https://journals.plos.org/plosone/s/recommended-repositories. You also have the option of uploading the data as Supporting Information files, but we would recommend depositing data directly to a data repository if possible.

Response: Please let us upload the data as Supporting Information files

Kind regards,

Reviewer #1:

1. Though miRNAs as biomarkers of postmenopausal osteoporosis have been studied intensively, certain miRNAs expressed in plasma at different stage of postmenopausal osteoporosis are still not clear. In current study, plasma miRNA-21-5p and miRNA-422a were chosen to verify their association with osteoporosis in postmenopausal women. Conclusion: “these miRNAs' expression levels recommend as helpful markers for screening women at risk of osteoporosis.” Which is more helpful or both?

Response: Thanks a lot for your attention to our submitted manuscript. In our study, the expression of both miRNAs in postmenopausal women with osteoporosis is higher than normal. miRNA-21 and miRNA-422a had good and excellent diagnostic accuracy, respectively. Diagnostic accuracy of BTMs was explained in abstract and main text. Figure 3 was added in this regard.

Method and Materials Participants: “They were referred to the Bone Densitometry center. Based on densitometric tests, 142 normal- 85 BMD, 109 osteoporosis, and 194 osteopenia women were identified.” It is suggested that the expression of miRNA-21-5p and miRNA-422a should be detected in some women qualified the inclusion criteria of 194 osteopenia women. Thus, the candidate miRNAs as biomarkers of postmenopausal patients in the early stage may be further explored through the comparison of osteoporosis, osteopenia, and normal groups, and maybe contribute to predict the occurrence of postmenopausal osteoporosis.

Response: Thank a lot for your suggestion.

2. Potential target genes of miRNA-21-5p and miRNA-422a can be predicted through bioinformatic analysis and subsequently verified.

Response: Thank a lot for your attention to our submitted manuscript. This was added to the suggestions at the end of discussion.

3. The primer sequences used for qRT-PCR should be listed as attached table.

Response: The primer sequences used for qRT-PCR were listed in table 1.

4. Please standardize the statistical symbol p as lowercase and italic type.

Response: statistical symbol p changed to lowercase and italic type. 

Reviewer #2: In this manuscript, the authors have investigated the association between plasma miRNA-21-5p and miRNA-422a with osteoporosis in postmenopausal women. By evaluating 65 osteoporotic and 61 normal cases of women aged 50-65, it has been concluded that miRNA-21 and miRNA-422 are significantly upregulated in osteoporotic women compared to normal. The results further reveal that there are inverse correlations between the levels of miRNA-21 and miRNA-422a and bone mineral density of spine and femoral neck in osteoporotic samples. The role of miRNA-422a and miRNA21 have been investigated in postmenopausal osteoporosis. For example, miRNA 422a has been studied in osteoclast precursor cells as a biomarker and miRNA 21 has been shown to regulate bone turn over markers. This investigation using plasma miRNAs further confirms that miRNAs 422a and 21 could be explored as diagnostic markers in postmenopausal osteoporosis.

Minor comment:

1. Although, the results are quite convincing and interesting, the sample size (65 osteoporotic and 61 normal) is too small.

Response: Thank you. We referred to it in limitations. 

2. The manuscript has few typographical errors that need to be fixed.

Response: Typographical errors were solved as possible. Thank a lot for your attention to our submitted manuscript.

Regards

---

## [Editor Report · Decision Letter 1]

6 Jun 2023

Up-regulation of plasma miRNA-21 and miRNA-422a in postmenopausal osteoporosis

PONE-D-22-34068R1

Dear Dr. Farshbaf-Khalili,

We’re pleased to inform you that your manuscript has been judged scientifically suitable for publication and will be formally accepted for publication once it meets all outstanding technical requirements.

Kind regards,

Gary S. Stein

Academic Editor

PLOS ONE
---

## [Editor Report · Acceptance letter]

13 Jun 2023

PONE-D-22-34068R1 

Up-regulation of plasma miRNA-21 and miRNA-422a in postmenopausal osteoporosis 

Dear Dr. Farshbaf-Khalili:

I'm pleased to inform you that your manuscript has been deemed suitable for publication in PLOS ONE. Congratulations! Your manuscript is now with our production department. 

Kind regards, 

on behalf of

Dr. Gary S. Stein 

Academic Editor

PLOS ONE